# *Acacia catechu* Willd. Extract Protects Neuronal Cells from Oxidative Stress-Induced Damage

**DOI:** 10.3390/antiox11010081

**Published:** 2021-12-29

**Authors:** Elda Chiaino, Roberto Stella, Caterina Peggion, Matteo Micucci, Roberta Budriesi, Laura Beatrice Mattioli, Carla Marzetti, Federica Pessina, Massimo Valoti, Maria Frosini

**Affiliations:** 1Dipartimento di Scienze della Vita, Università di Siena, Via Aldo Moro 2, 53100 Siena, Italy; chiaino@student.unisi.it (E.C.); massimo.valoti@unisi.it (M.V.); 2Dipartimento di Chimica, Istituto Zooprofilattico Sperimentale delle Venezie, Viale Dell’Università 10, 35020 Legnaro, Italy; rstella@izsvenezie.it; 3Dipartimento di Scienze Biomediche, Università di Padova, Via Ugo Bassi 58/b, 35131 Padova, Italy; caterina.peggion@unipd.it; 4Dipartimento di Farmacia e Biotecnologie, Alma Mater Studiorum-Università di Bologna, Via Belmeloro 6, 40126 Bologna, Italy; matteo.micucci2@unibo.it (M.M.); roberta.budriesi@unibo.it (R.B.); laura.mattioli13@unibo.it (L.B.M.); 5UniCamillus-Saint Camillus International University of Health Sciences, Via di Sant’Alessandro 8, 00131 Rome, Italy; 6Valsambro S.r.l., Via Cairoli 2, 40121 Bologna, Italy; carla.marzetti@valsambro.it; 7Dipartimento di Medicina Molecolare e dello Sviluppo, Università di Siena, Via Aldo Moro 2, 53100 Siena, Italy; federica.pessina@unisi.it

**Keywords:** *Acacia catechu* Willd., neuroprotection, neurodegenerative diseases, oxidative stress, brain slices, SH-SY5Y cells, proteomic, apoptosis, caspase, CaMKII

## Abstract

Oxidative stress (OS) and the resulting reactive oxygen species (ROS) generation and inflammation play a pivotal role in the neuronal loss occurring during the onset of neurodegenerative diseases. Therefore, promising future drugs that would prevent or slow down the progression of neurodegeneration should possess potent radical-scavenging activity. *Acacia catechu* Willd. heartwood extract (AC), already characterized for its high catechin content, is endowed with antioxidant properties. The aim of the present study was to assess AC neuroprotection in both human neuroblastoma SH-SY5Y cells and rat brain slices treated with hydrogen peroxide. In SH-SY5Y cells, AC prevented a decrease in viability, as well as an increase in sub-diploid-, DAPI positive cells, reduced ROS formation, and recovered the mitochondrial potential and caspase-3 activation. AC related neuroprotective effects also occurred in rat brain slices as a reversal prevention in the expression of the main proteins involved in apoptosis and signalling pathways related to calcium homeostasis following OS-mediated injury. Additionally, unbiased quantitative mass spectrometry allowed for assessing that AC partially prevented the hydrogen peroxide-induced altered proteome, including proteins belonging to the synaptic vesicle fusion apparatus. In conclusion, the present results suggest the possibility of AC as a nutraceutical useful in preventing neurodegenerative diseases.

## 1. Introduction

Neurodegenerative diseases (ND) are age-dependent disorders, the prevalence of which has been increasing in the last decades, owing mainly to the rapidly growing proportion of the elderly population [1]. These diseases are diverse in their pathophysiology, commonly causing memory and cognitive impairments, or affecting movements. Despite their clinical heterogeneity, ND share many common hallmarks in their aetiology, such as the accumulation of misfolded proteins, which occurs alongside neuronal loss in specific brain areas [2]. High levels of oxidative stress (OS) are usually described in the brain of patients suffering from ND, and an overload of reactive oxygen species (ROS) is reported to induce mitochondrial function impairment, activate apoptotic cascades and neuroinflammation, and contribute to ND onset and progression via both redox imbalance and altered cellular signalling [3]. Disrupted calcium signalling may promote ND-associated proteins misfolding and aggregation and these, in turn, might affect calcium signalling itself, thus compromising the ability of neurons to respond to excitotoxic challenges [4]. The key players in such a pathological interplay seem to be molecules present in excitatory synapses that are downstream of calcium signalling, like the calcium/calmodulin (CaM)-dependent protein kinase II (CaMKII) [4].

Owing to the growing number of cases, the search for new approaches for preventing or treating ND is intensive, but an efficacious therapy has not been outlined yet. Preventing the progression of neurodegeneration is fundamental, as in chronic ND, the pathological changes at cellular levels precipitate the clinical onset of the disease by several years. For this, hampering ROS formation may be a promising approach, and natural products may offer great potential from a clinical point of view [5,6].

*Acacia catechu* Willd. extract has been largely used for the treatment of several diseases especially in traditional medicine thanks to its hepatoprotective, antipyretic, antidiarrheal, hypoglycaemic, anti-inflammatory, immunomodulatory, antinociceptive, spasmolytic, and antispastic properties [7,8]. *Acacia catechu* Willd. contains high amounts of flavonoids, such as flavan-3-ols, (+)-catechin, (−)-epicatechin, (−)-epicatechin-3-*O*-gallate, and (−)-epigallocatechin-3-*O*-gallate [9]. These derivatives and related polyphenols possess antioxidant, anti-inflammatory, and anti-apoptotic effects and most of them have been shown to be neuroprotective against neural injuries and ND [10].

The aim of the present work was to assess the neuroprotective properties of a preparation obtained by *Acacia catechu* Willd. heartwood by decoction (AC), recently characterized for its high catechin and epicatechin content [11], toward OS-mediated injury. Human neuroblastoma SH-SY5Y cells treated with hydrogen peroxide were firstly used, and the encouraging results drove the assessment of neuroprotection in rat brain slices, an experimental model that largely preserves the tissue architecture of the brain regions from which they originated. This maintains neuronal activities with intact functional local synaptic circuitry, thus allowing a better extrapolation of the results in terms of neuroprotection [12]. In brain slices, the expression of the main proteins involved in apoptosis and signalling pathways related to calcium homeostasis following OS-mediated injury, along with the effects of AC, were investigated. Additionally, a deep shotgun proteomics study based on label-free quantification proteomic analysis by liquid chromatography coupled with high-resolution tandem mass spectrometry (LC-HRMS/MS) was performed to unveil OS-induced changes in the protein expression pattern of brain slices and whether AC neuroprotective effects positively affected it.

## 2. Materials and Methods

### 2.1. Plant Materials

*Acacia catechu* Willd. heartwood by decoction (AC) was supplied by BIO-LOGICA S.R.L. (Via della Zecca 1, 40100 Bologna, Italy). The hydroalcoholic fluid extract production from the plant was obtained by maceration and percolation according to the European Pharmacopoeia 8.0. For more insight, please visit Minardi (A. Minardi and figli s.r.l. via Boncellino 18/A 48012 Bagnacavallo (RA) Italy; website: www.minardierbe.it (accessed on 26 October 2021)). AC was characterized for its catechins content, by using an HPLC coupled with UV detection [11], and (±)-Catechin and (−)-Epicatechin were found to be 31.5 ± 0.82 mg/g and 12.5 ± 0.42 mg/g, respectively. Further characterization of the sample was carried out by means of a chiral method based on cyclodextrin-modified micellar electrokinetic chromatography (CD-MEKC) and the presence of both the enantiomers, namely, the native (+)-Catechin and the artifact (−)-Catechin, at approximately the same content level was found [11]. The latter was assumed as a marker of the epimerization as a consequence of the process applied in decoction preparation (thermal treatment). Analysis of decoction preparations stored during a very long-term period (at least two years) at room temperature in the dark, did not show loss of (−)-EC and (+)-C, nor epimerization progression, thus suggesting very high chemical stability of the preparation.

*Acacia catechu* Willd. hydroalcoholic heartwood fluid extract (AC) (BIO-LOGICA S.R.L., Bologna, Italy) was obtained according to the European Pharmacopoeia 8.0. by maceration and percolation (for more details see www.minardierbe.it (accessed on 26 October 2021)). An HPLC coupled with UV detection AC was used for catechins quantitative analysis [11], finding (±)-Catechin (31.5 ± 0.82 mg/g) and (−)-Epicatechin (12.5 ± 0.42 mg/g) as the most representative. More details are given in [11].

### 2.2. Neuroprotection Assessed on Human Neuroblastoma SH-SY5Y Cells

#### 2.2.1. Cell Cultures and Hydrogen Peroxide-Induced Injury

Human SH-SY5Y neuroblastoma cells (ECACC Cat# 94030304, passages 6–20), were cultured according to standard conditions, as already reported [13,14]. To induce OS, SH-SY5Y cells were treated with H_2_O_2_ (25 µM freshly prepared, 1 h), followed by 24 h with medium [14,15].

#### 2.2.2. AC Treatments

AC stock solution (10 mg/mL in PBS, pH adjusted to 7.3), prepared nearly before use, was carefully filtered (0.45 µm pore size) prior to dilution to the desired final concentration with cell culture medium. In chronic ND, beginning pharmacological treatments before the occurrence of clinical symptoms is crucial in order to stop or slow pathological changes at cellular levels, which precipitate the clinical onset of the disease by several years. As preventing the progression of neurodegeneration is fundamental, and in order to better extrapolate the results, the potential of AC in preventing OS-mediated injury was assessed by treating SH-SY5Y cells with the extract (1–10 μg/mL) 2 h before (pre-treatment protocol), or 2 h before and during cell injury (pre- and co-treatment) [15] according to the scheme reported in Figure 1.

#### 2.2.3. Cell Viability Assays

To provide information about the SH-SY5Y cells metabolic activity (viability), the 3-(4,5-Dimethylthiazol-2-yl)-2,5-diphenyl-tetrazolium bromide (MTT) assay was performed. Cytotoxicity was related to the decline in absorbance and medium-treaded samples (controls, 100% viability) were compared vs. treated ones [15,16].

#### 2.2.4. Cell Cycle Analysis

Cell cycle and apoptotic, sub-G0/G1 population analysis were performed using flow cytometry according to protocols already reported. SH-SY5Y cells were used according to protocols already reported [11,13,14,15]. A FACScan flow cytometer coupled with Cell Quest software v. 3.0 (BD Biosciences, San Jose, CA, USA) was used to detect red fluorescence (DNA, FL2 channel, 10^4^ cells/sample) and to calculate the percentage of cells in the different phases of the cell cycle.

#### 2.2.5. Fluorescence Microscopy Assays: DAPI and Rhodamine-123 Staining

Assessment of apoptosis based on nuclear morphology was performed using the 4′,6-diamidino-2-phenylindole (DAPI) staining kit (Life Technologies Italia, Monza, Italy) according to protocols already established [14,15,17,18]. Mitochondrial integrity was assessed by using a rhodamine-123-based assay [11,18].

#### 2.2.6. Intracellular ROS Content and Caspase-3 Activity

The fluorescence probe 2′,7′-dichlorofluorescein diacetate (DCFH-DA, Sigma Aldrich, St. Louis, MO, USA) was used to detect intracellular ROS at the end of the treatments [11,13,14]. The caspase-3 assay was performed using the specific substrate DEVD-AMC (Ac-Asp-Glu-Val-Asp-7-amino-4-methylcoumarin), which releases the fluorescence dye 7-amino-4-methylcoumarin (AMC, 380 nm excitation and 460 nm emission) upon caspase-3 activation [11,14].

### 2.3. Neuroprotection on Rat Brain Slices

#### 2.3.1. Slices Preparation

The experimental protocol, already described in [14,19,20,21] is summarized in Figure 2. Briefly, male Wistar rats cortical slices (400 μm, manual chopper Stoelting Co., Wood Dale, IL, USA) were placed in artificial cerebrospinal fluid (ACSF, composition in mM: 120 NaCl; 2 KCl; 1 CaCl_2_; 1 MgSO_4_; 25 HEPES; 1 KH_2_PO_4_; and 10 glucose, final pH 7.4, bubbled with a 95% O_2_, 5% CO_2_ gas mixture) and left for 30 min at ambient temperature to recover from trauma (equilibration phase 1). Then, the slices were transferred into 24-well culture plates (2–3 slices/well, mean weight ~30–40 mg, in 0.5 mL ACSF at 37 °C) and left for 60 min (equilibration phase 2). During this period, the medium was removed and replaced with fresh, oxygenated ACSF every 15 min. After equilibration phase 2, slices were pre-treated with AC for 60 min, which was then followed by hydrogen peroxide injury (5 mM for 60 min, which causes about 50% tissue death [14,15,20]). Hydrogen peroxide was freshly prepared from a 30% stock solution prior to each experiment.

#### 2.3.2. AC Treatment

In agreement with results obtained with SH-SY5Y cells and to reduce as much as possible the number of animals used in the study, AC treatment was performed according to the pre- and co-treatment protocol. After equilibration phase 2, slices were incubated with ACSF containing (or not, in the case of controls) AC (1–200 µg/mL in ACSF) for 1 h. After this period, the medium with AC was maintained and H_2_O_2_ (5 mM for 1 h) was added. At the end of the treatments, the slices were used for assessing tissue viability, ROS, and MDA levels or treated as reported below for Western blot (WB) and proteomic analysis.

#### 2.3.3. Viability Assays

The colorimetric MTT 3-(4,5-dimethylthiazol-2-yl)-2,5-diphenyltetrazolium bromide method was used to assess tissue viability [14,19,20]. At the end of the treatments, slices were washed with ACSF and incubated with 0.5 mg/mL of MTT (Sigma Aldrich, St. Louis, MO, USA, 300 µL/well) at 37 °C for 45 min in the dark. Then, slices were gently transferred into a 96 multi-well plate and 200 µL of DMSO was added and incubated for 30 min at 37 °C in a shaking plate. Afterward, 100 µL of supernatants were collected and the formazan product was measured at 560 and 630 nm (OD560–OD630, Multiskan TM GO, Thermo Scientific, Waltham, MA, USA). In some experiments, the slices were immersed in 2 mL of 4% formalin for 24 h in the dark to fix the formazan, and after gently drying them with paper, photo images were taken [22]. The area of the injury, as identified by reduced MTT staining, was traced by an expert, blind to the treatment operator, calculated by using ImageJ software (National Institute of Health, Bethesda, MD, USA, 1.37v), and reported as a percent value of the total area of the slice.

#### 2.3.4. ROS and Lipid Peroxidation

Before H_2_O_2_ treatment, slices were loaded with ACSF containing DCFH-DA (20 µM, 10 min). At the end of the experiments, slices were carefully washed with cold PBS and homogenized in 500 µL of PBS. Fluorescence was measured using wavelengths of excitation and emission of 480 and 520 nm, respectively (Synergy HTX multi-mode reader, BioTek, Winooski, VT, USA), and normalized to the content of proteins. Average values in control slices were taken as 100% [23]. Lipid peroxidation was measured by the detection of thiobarbituric acid-reactive substances, according to previous reports [24].

#### 2.3.5. Brain Slices Lysis and WB Analyses

Brain slices were lysed in ice-cold buffer containing 8 M urea, 2% (*w*/*v*) SDS, 100 mM Tris-HCl, pH 8, and a protease inhibitors cocktail. To ensure complete lysis, samples were passed through an insulin syringe, incubated for 5 min in an ultrasound bath, and finally centrifuged (10 min, 14,000× *g*, 4 °C) to remove tissue debris. The homogenates were subjected to a Micro-Lowry assay (Sigma Aldrich, St. Louis, MO, USA) to determine total protein concentration. Samples were then stored at −80 °C until use.

For Western blot (WB) analyses, samples were diluted to an equal protein concentration using the loading sample buffer (62.5 mM Tris-HCl, pH 6.5, 2.3% *w/v* SDS, 10% *w/v* glycerol, 50 mM dithiothreitol (DTT), and bromophenol-blue 0.004% *w*/*v*). An amount of 15–20 µg of total proteins was subjected to SDS-polyacrylamide gel electrophoresis on precast gels (4–15% acrylamide/bis-acrylamide gradient concentration, Bio-Rad, Hercules, CA, USA), and then electroblotted onto polyvinylidene difluoride (PVDF) membranes (0.22 µm pore size, Bio-Rad, Hercules, CA, USA). Membranes were stained with Coomassie blue solution (0.1% (*w*/*v*) Brilliant Blu R (Sigma Aldrich, St. Louis, MO, USA), 50% (*v*/*v*) methanol, 7% (*v*/*v*) acetic acid) to verify the equal loading (for subsequent densitometric analyses, see below). After de-staining using methanol (100% *v*/*v*), membranes were blocked using TRIS-buffered saline solution (10 mM Tris-HCl, pH 7.5, 150 mM NaCl) added with 0.1% (*v*/*v*) Tween-20 (TBS-T) and 3% (*w*/*v*) bovine serum albumin (BSA) (1 h, room temperature), followed by addition of the desired primary antibody (overnight, 4 °C) diluted in TBS-T with 1% (*w*/*v*) BSA. After three washes (5 min. each in TBS-T), PVDF membranes were incubated with a horseradish peroxidase-conjugated anti-rabbit-IgG, anti-mouse IgG (Sigma Aldrich, St. Louis, MO, USA, cat. no. A0545 and A9044, respectively) or anti-goat (Santa Cruz Biotechnology, Dallas, TX, USA, cat. no. SC-2354) secondary antibody, depending on the primary antibody host specification. After the membranes were washed with TBS-T, immunoreactive bands were visualized and digitalized with the NineAlliance UVITEC imaging system (Eppendorf), using an enhanced-chemiluminescence reagent kit (Millipore, Burlington, MA, USA). For densitometric analysis, band intensities were evaluated with UVITEC image analysis software. Briefly, to determine relative protein abundance, the intensity of each immunoreactive band was normalized to the optical density of the corresponding Coomassie blue-stained lane [25,26]. The phosphorylation state of the proteins analysed was determined by calculating the ratio between the phosphorylated protein and the corresponding total protein immunoreactive band intensities.

The following primary antibodies were used (dilutions are in parentheses): rabbit anti-cleaved caspase-3 (1:500, Cell Signalling Technology, Danvers, MA, USA, cat. no. 9661), rabbit antio-caspase-3 (8G10) (1:500, Cell Signalling Technology, Danvers, MA, USA, cat. no. 9665), rabbit anti-Bax (1:500, Santa Cruz Biotechnology, Dallas, TX, USA, cat. no. SC6236); rabbit anti-Bcl2 (1:1000, Sigma-Aldrich, cat.no. SAB4500003), rabbit anti-phosphorylated (on Thr 286) CAMKII (1:500, Cell Signaling Tecnology cat no. 12716), rabbit anti-CAMKII (1:1000, Cell Signaling Technology, cat. no. 3362), rabbit anti-ERK1/2 (1:1000, Cell Signaling Technology, cat. no. 9102), mouse anti-phosphorylated (on Thr202/Tyr204) ERK 1/2 (pERK1/2, 1:1000, Cell Signaling Technology cat. no. 9106), rabbit anti-p38 (1:1000, Cell Signaling Technology; cat. no. 9212), rabbit anti-phosphorylated (on Thr180/Tyr182)-p38 (p-p38, 1:1000, Cell Signaling Technology, cat. no. 9211), goat anti-SERCA2a (1:1000, Santa cruz biotechnology; cat no. sc-8094).

#### 2.3.6. Proteomics Analysis

Protein extracts from 4 independent rat brains undergoing the treatments described above (control, H_2_O_2_ and H_2_O_2_ + AC 200 µg/mL) were processed according to the filter-aided sample preparation protocol [27]. Additionally, three independent quality control (QC) samples prepared by mixing an equal amount of each protein extract were prepared.

Equal amounts of each protein extract (200 μg in 50 μL) were mixed with 150 μL of 8 M urea in 100 mM Tris-HCl, pH 8. Samples were loaded in filters with a molecular weight cut-off of 10 kDa (Sartorius, Goettingen, Germany) and centrifuged (15 min, 14,000× *g*, 25 °C). Then, three washing steps were performed by adding 200 μL of 8 M urea in 100 mM Tris-HCl, pH 8, and by centrifuging samples (15 min, 14,000× *g*, 25 °C). Afterward, disulphide bonds were first reduced using 200 μL of 25 mM DTT (1 h, 55 °C) and, after centrifugation, cysteine residues were alkylated using 200 μL of 55 mM iodoacetamide (45 min in the dark, RT) (both diluted in 100 mM Tris-HCl, pH 8). After centrifugation, five additional washing steps were performed by adding 100 μL of 100 mM Tris-HCl, pH 8, to each filter. Finally, protein digestion was carried out by incubating filters (18 h, 37 °C) in the presence of 100 μL of 100 mM Tris-HCl, pH 8, and 1 mM CaCl_2_, containing 8 μg of sequencing grade modified trypsin (enzyme: protein ratio of 1:25 (*w*/*w*)).

After digestion, released peptides were collected by centrifugation (15 min, 14,000× *g*). To each filter, an additional amount of 100 μL of 100 mM Tris-HCl, pH 8, was added to improve the recovery of peptides by a second centrifugation of 15 min, 14,000× *g*.

Protein digests were acidified by adding formic acid until a pH ≤ 3 was reached, and then desalted by using C_18_ BioPure spin columns (The Nest Group, Southborough, MA, USA), following the manufacturer’s instructions. Briefly, columns were activated with 400 μL of methanol, and then with 400 μL of acetonitrile. Then, columns were equilibrated twice with 400 μL of water containing formic acid (0.1%, *v*/*v*). Then, peptides were loaded onto C_18_ spin columns and washed twice with 200 μL of water containing formic acid (0.1%, *v*/*v*). The elution was achieved by adding 200 μL of 80% acetonitrile in water (*v*/*v*) containing 0.1% formic acid (*v*/*v*). Peptide extracts were finally dried at room temperature under a stream of nitrogen and dissolved in 200 μL of 5% acetonitrile in water (*v*/*v*) containing 0.1% formic acid (*v*/*v*), immediately before LC-HRMS/MS analyses (protein digest concentration of 1 μg/μL).

Label-free quantification of proteins was performed by analysing digested peptides with a Q-Exactive mass spectrometer, coupled with a UHPLC system Ultimate 3000 (Thermo Fisher Scientific, Germany). Peptides were separated on a reversed phase analytical column (Aeris peptide C_18_, 150 × 2.1 mm, 2.6 μm, Phenomenex, USA). Mobile phases were water (A) and acetonitrile (B), both containing 0.1% formic acid (*v*/*v*). The gradient applied was: 0–1 min 2.5% B, then B was increased linearly to reach 30% at 20 min and increased again to reach 50% at 24 min. Afterward, solvent B was increased to 95% at 26 min and kept until 30 min to wash the column. Finally, the concentration of B was decreased to 2.5% at 30.5 min, to equilibrate the column to initial conditions until 35 min. The column oven was set to 30 °C, the injection volume was 5 µL and the flow rate was 200 μL/min.

Full scan HRMS and fragmentation MS/MS spectra were acquired in positive ionization mode using the following source parameters: capillary temperature 325 °C, heater temperature 325 °C, sheath gas flow rate 35 a.u. (arbitrary units), auxiliary gas flow rate 10 a.u., spray voltage 3 kV, and S-lens voltage 55 V. Full HRMS scans were acquired at 70,000 resolution full width at half maximum (FWHM) followed by the MS/MS scan of the four most intense precursor ions acquired at 17,500 resolution FWHM. Full scan spectra were acquired in a scan range from 300 to 2000 Th. Higher energy C-trap dissociation (HCD) fragmentation was performed with a normalized collision energy (NCE) of 27, and a dynamic exclusion of 30 s. Each protein digest was analysed twice by LC-HRMS/MS analysis. In the second run, an exclusion list containing the *m/z* values deriving from peptides identified in the first run was applied to increase the number of identified peptides and increase the overlap among different samples.

Three QC samples were used to ensure repeatability of protein quantification and to check for instrumental performance along with the analytical session.

#### 2.3.7. Database Search and Label-Free Quantification

LC-HRMS/MS data were processed using Proteome Discoverer (Thermo Fischer Scientific, version 2.1) and analysed with SEQUEST HT (Thermo Fisher Scientific). Protein identification was performed using the following parameters: enzyme specificity set to trypsin with up to 1 allowed miss cleavage, peptide tolerance set to 10 ppm, and fragment mass tolerance set to 0.2 Da.

Searches were performed against the *Rattus norvegicus* reference proteome of UniProtKB (version UP000002494). Carbamidomethyl-Cys was set as a fixed modification, while oxidation of Met, pyro-Glu, and acetylation of the *N*-terminus was set as a variable modification. Proteins were considered as positively identified if at least two independent peptides were identified with a false discovery rate below 5%. Proteins were grouped into protein families according to the principle of maximum parsimony. Label-free quantification was achieved by using the precursor ion abundance of unique peptides that are not shared between different proteins or protein groups and normalizing the relative quantification values to the total peptide amount. Normalized intensity values of proteins derived from the Proteome Discoverer software were finally exported in a spreadsheet, for multivariate analysis.

### 2.4. Analysis of Data

Results were reported as means ± SEMs of at least four (SH-SY5Y cells) or three-five (brain slices) independent experiments. Statistical significance was assessed by using a one-way ANOVA followed by Bonferroni or by Holm Sidak’s post-hoc tests, as appropriate (GraphPad Prism version 5.04, GraphPad Software Inc., San Diego, CA, USA). In all comparisons, the level of statistical significance (*p*) was set at 0.05. Proteomics data were mean-centered and scaled to unit variance prior to analysis by partial least square discriminant analysis (PLS-DA) using SIMCA-P software (version 13.0).

## 3. Results

### 3.1. AC Prevented OS-Induced Decrease in SH-SY5Y Cells Viability

The potential of AC to prevent OS-mediated injury was initially assessed according to the pre-treatment protocol. This approach, however, revealed to be ineffective as the drop in cell viability caused by the H_2_O_2_ challenge remained mostly unchanged despite the pre-treatment with the extract (see Appendix A). Interestingly, however, when AC was used as a pre- and co-treatment, SH-SY5Y cells were more resistant to the injury. In fact, the viability, was significantly recovered by ~26.23 and 37.7% for 1 and 10 µg/mL AC, respectively (*p* < 0.01 vs. H_2_O_2_) (Figure 3a). Therefore, the pre-and co-treatment protocol was then selected to further assess the neuroprotective effects of AC. Finally, the extract per se was inactive, as after 24 h treatment with 10 µg/mL, SH-SY5Y cell viability was comparable to that of the control slices.

### 3.2. AC Prevented the Formation of ROS and the Loss in Mitochondria Membrane Potential Caused by OS

As reported in Figure 3, panel b, AC lowered in a concentration-dependent fashion the intracellular ROS formation caused by H_2_O_2_, with a maximum effect achieved at 10 µg/mL (−71.9 ± 3.3%, *p* < 0.01 vs. H_2_O_2_). The mitochondrial membrane potential (Ψm) is closely linked to functional activity and loss in Ψm (depolarization) is an initial sign of apoptosis, being a result of mitochondrial uncoupling [28]. Thus, R123 staining was used to check for changes in Ψm.

In healthy cells, the dye was sequestered by active mitochondria according to the negative membrane potential across the inner membrane, thus leading to high green fluorescence. On the contrary, H_2_O_2_-treated cells presented a weaker green fluorescence intensity because of the loss in mitochondrial membrane potential and, as a consequence of the dye, an effect which was prevented by AC (10 µg/mL) (Figure 3, lower panels).

### 3.3. AC Reduced Apoptotic-Mediated SH-SY5Y Cell Death Caused by OS

Cell cycle analysis showed that after OS, the percentage of sub G0/G1 hypodiploid, apoptotic cells was significantly higher (~12.0%, *p* < 0.001 vs. untreated cells) (Figure 4a). Interestingly, flow cytometry data also indicated that AC exerted neuroprotection, as the percentage of cells in sub G0/G1 were gradually reduced upon increasing AC concentration. Finally, the number of cells in the G0/G1 and S phase were mostly unchanged in both OS and OS + AC conditions, while those in the G2/M were decreased after OS, but recovered basal value after the treatment with the extract (Figure 4b). Nuclear apoptotic changes, assessed by using the fluorescent dye DAPI, confirmed previous results. In untreated SH-SY5Y cells, few of them were with fragmented nuclei and condensed DNA, at variance with OS, in which a high number of cells showed these characteristics. In the presence of AC, however, a reduced number of nuclei stained with DAPI occurred and cells showed nucleus shape and nucleus staining intensity comparable to controls (Figure 4, lower panels).

### 3.4. The Increase in Caspase-3 Activity Caused by OS Was Reduced by AC

OS triggered by ROS activates a series of signalling events ultimately leading to programmed cell death, or apoptosis. Caspase-3 represents a convergence point for both mitochondria-dependent and -independent pathways in cells undergoing apoptotic cell death in response to OS [28]. To investigate whether AC could prevent the activation of apoptosis pathways caused by OS, a specific fluorescent caspase-3 substrate releasing the fluorescence probe AMC when activated was used, and 10 µg/mL AC was tested as the most active neuroprotective concentration in the previous assays. Results showed that H_2_O_2_ challenge doubled the release of the fluorescent substrate (Figure 4c), but this effect was completely prevented by AC, as proven by the regained basal values of AMC-derived fluorescence.

### 3.5. AC Neuroprotection Occurred Also in Rat Brain Slices Subjected to OS

Results on SH-SY5Y highlighted the neuroprotective effects of AC towards OS-induced injury. AC effects were thus assessed in a tissue context such as rat brain slices, in which the main structural and synaptic organization of the original tissue is conserved [12]. For these experiments, AC pre-treatment (1–200 µg/mL, 1 h) was performed, which was followed by an OS challenge (H_2_O_2_ 5 mM, 1 h) still in the presence of AC, assessing at the end slice viability. OS caused a significant tissue injury (~40.0%, *p* < 0.001 vs. CTRL) (Figure 5a), which was confirmed by slices image analysis (see Appendix A). Interestingly, AC prevented the effects of OS at 100 and 200 µg/mL, while 10 and 50 µg/mL were ineffective. Finally, 2 h of treatment with AC 200 µg/mL, did not affect brain slice viability.

### 3.6. AC Reverted ROS and Lipid Peroxidation in Brain Slices Caused by OS

As previously mentioned, OS induces the formation of ROS, which in turn can react with the polyunsaturated fatty acids of lipid membranes causing lipid peroxidation [28]. Thus, the occurrence of both was assessed in brain slices treated with H_2_O_2_, along with the ability of AC to prevent their formation. AC was used at the concentrations that resulted in being the most effective in the MTT assay (100 and 200 µg/mL). As reported in Figure 5b,c, OS caused a huge increase in both ROS and MDA formation, which was, however, completely prevented by AC.

### 3.7. AC Prevented Apoptotic Activation in Rat Brain Slices

To assess apoptotic cell death activation, along with the neuroprotective activity of AC, treated brain slices were subjected to WB analyses using specific antibodies to both the inactive procaspase 3 and its cleaved activated isoform, to B-cell lymphoma2-associated X protein (Bax, proapoptotic) and to B-cell lymphoma-2 (Bcl-2, antiapoptotic) proteins. Changes in the Bax/Bcl-2 ratio (i.e., index of the apoptotic potential of a cell [28] was evaluated, and AC effects were assessed by using its most effective concentration of 200 µg/mL. As shown in Figure 6, AC prevented the activation of caspase-3 and the up-regulated Bax/Bcl-2 ratio, known to be altered by H_2_O_2_ treatment in SH-SY5Y cells [29] as well as in in vivo experimental models of ND [30].

Deleterious effects of OS on rat brain slices was also confirmed by monitoring the protein level of the sarco-endoplasmic reticulum Ca^2+^ ATPase isoform 2 (SERCA2a, the SERCA isoform present in the nervous system [31], whose expression was already reported to be impinged by a high amount of ROS in different models [32,33]. Results showed that SERCA2a was significantly downregulated in H_2_O_2_-treated rat brain slices compared to those not treated (control). Unfortunately, AC was not effective in recovering SERCA2a level/functionality (Appendix A), suggesting that the ER-residing Ca^2+^ pump is not the primary target and is not involved in AC-induced neuroprotective effects.

### 3.8. AC Protects against the CAMKII Activation in Rat Brain Slices

As already reported, one of the main consequences of OS is the activation of CaMKII pathways that, in turn, triggers the activation of several signalling pathways, including the proteins extracellular signal-regulated kinase 1 and 2 (ERK 1/2) and the mitogen-activated protein kinase (MAPK) p38 [34,35]. Thus, the phosphorylation of the above-mentioned proteins was assessed. As reported in Figure 7, the H_2_O_2_ challenge caused a significant increase in CaMKII and ERK 1/2 activation, at variance with p38 which was unaffected. Interestingly, AC 200 µg/mL completely prevented the phosphorylation/activation of CaMKII, while only a tendency toward a diminished activation of ERK1/2 was observed.

### 3.9. OS-Induced Proteomics Changes Are Reverted by AC in Rat Brain Slices

The changes in CaMKII- and ERK1/2-related signalling pathways, prompted us to investigate whether these could affect the proteome of rat brain slices and whether the AC beneficial effects may be linked to the prevention of such changes. This task was accomplished by comparing the proteomic pattern of rat brain slices treated with ACSF (controls), subjected to OS-induced injury (H_2_O_2_), or to OS-induced injury in the presence of AC 200 µg/mL (H_2_O_2_ + AC) thanks to a shotgun proteomics approach exploiting label-free quantification. The adopted experimental procedure allowed us to identify 8297 peptides coming from 839 proteins. Of these proteins, 665 were correctly identified and quantified in at least three out of four brain samples coming from the different tested conditions. The overall repeatability of the adopted sample preparation procedure was attested by the mean CV% value calculated from QC samples that resulted in 17.7%, attesting for the absence of significant instrumental drift during the analytical sessions.

To highlight changes in the proteomics profile caused by OS-induced injury, a PLS-DA was performed by comparing ACSF- to H_2_O_2_-treated slices. This supervised analysis suggested that in the latter case, changes in the protein expression profile occurred, as attested by the clear separation of the two sample groups (Figure 8a).

Characteristic parameters of the PLS-DA model showed that the three principal components were sufficient to explain more than 90% of data variance (R^2^ value) and that the predictive capability of such a model (Q^2^ value) is high, thus attesting for the good fitting of the model itself (Figure 8b). Proteins driving the separation of sample groups in the PLS-DA model were selected by considering the variable importance in the projection (VIP value) and the CV% value calculated in QC samples for these proteins. Twenty-eight proteins possessing a VIP value above 1.5 and a CV% value calculated in QC samples below 30% were found. Such proteins were highlighted in red in the loading scatter plot of the PLS-DA and were found to be inversely or directly correlated to the OS-induced injury (Figure 8c). Finally, the capability of AC treatment to prevent the effects of OS on the proteome of rat brain slices was assessed by applying a second PLS-DA on the three different conditions. In accordance with the previous analysis, brain slices treated with H_2_O_2_ were clearly separated from control samples (ACSF-treated), while ACSF- and H_2_O_2_ + AC-treated brain slices were clustered together in the PLS-DA score plot, suggesting that no global changes occurred in the protein amount when AC was present (Figure 8 panel d). These findings were fully in agreement with previous observations attesting for neuroprotective effects of AC against OS-induced brain injury. Interestingly, 23 of the 28 proteins driving the separation between H_2_O_2_- and ACSF-treated brain slices in the PLS-DA plot were significantly affected by OS-induced brain injury (*p* < 0.05), and 18 of these were indeed rescued by AC treatment. Major details regarding proteins whose expression was altered by OS-induced brain injury are reported in Appendix A. Gene Ontology enrichment analysis (performed using the EnrichR webtool) evidenced that the protein component of exocytic and synaptic vesicles (such as Syngr3, Syn1, PCMT1, and SPTAN1), known to be also involved in the synaptic transmission process, were significantly enriched (Appendix A). This result is in agreement with our observation regarding the altered functionality of CaMKII, which controls the synaptic strength and plasticity (e.g., both through the phosphorylation of membrane receptors or by acting on factors that regulate, the transcription of molecules involved in such processes) [36,37].

## 4. Discussion

Remarkable advances in the understanding of ND are unfortunately paralleled by scarce success in finding effective therapies to replace those currently available, as the latter are mostly inadequate being not able to stop the progress of the disease. The attention towards the use of natural compounds is growing, in that these might have great potential in the prevention/treatment of ND [38]. Catechins have been reported to possess potent iron-chelating, radical-scavenging and anti-inflammatory activities and to exert neuroprotection in a wide array of cellular and animal models of neurological diseases [39,40]. In addition, catechins can modulate many signal transduction pathways, cell survival/death genes, and mitochondrial function [40], which significantly contribute to their overall activity. *Acacia catechu* Willd. heartwood extract used in the present study contains ~42 mg/g of catechins, mainly (±)-catechin hydrate and (−)-epicatechin [11], an amount higher than that reported in green tea [41], whose mean contents of (−)-EC and (±)-C is 6 mg/g and 1.5 mg/g, respectively, (i.e., up to ~10–20 times lower), as determined in almost one hundred of samples [42]. Furthermore, AC bark extract’s catechin content is less subjected to seasonal variation with respect to that in green tea or some fruits [43,44], and this makes AC a very interesting source of active polyphenols. Owing to these characteristics, the aim of the present study was to assess the neuroprotective properties of AC on both human neuroblastoma SH-SY5Y cells and in rat brain slices. The OS-induced injury was chosen as it is an important pathological culprit in ND, either as a triggering factor or as a crucial step of the downstream cascade which leads to neuronal death [3]. The present findings demonstrated that OS caused by H_2_O_2_ challenge induced ROS production and mitochondrial dysfunction accompanied by apoptotic-mediated SH-SY5Y cells death, in agreement with other reports on the same cell line [14,45,46,47]. To evaluate the potential of AC to prevent the injury, SH-SY5Y cells were treated with the extract 2h before (pre-treatment protocol) or 2h before and during cell injury (pre- and co-treatment protocol). Interestingly, AC prevented OS-mediated SH-SY5Ycell death, the formation of ROS, the loss in mitochondria membrane potential as well as caspase-3 activation. AC protective and antioxidant effects occurred in the range of 1–10 µg/mL, which correspond to ~0.1–1.5 µM catechin, highlighting a very high profile of activity of the extract. These polyphenols, in fact, especially (-)-Epigallocatechin-3-gallate (EGCG), protect SH-SY5Y cells at concentrations of at least one or two orders of magnitude higher [48,49]. The same compounds have an hormetic, bell shaped behaviour, characterized by an “efficacy window” of neuroprotective activity in the low micromolar range, whereas they become pro-oxidant and/or pro-apoptotic for higher (>10–50 µM) concentrations [50]. This is also the case of AC, which exerts pro-oxidant activity at 250 µg/mL in both HT-29 [11] and SH-SY5Y cells (unpublished observation), a concentration 25 times higher than that demonstrated to protect neurons. The wider limit between protectant- and damaging concentrations constitutes an added value of AC.

The results on human neuroblastoma cells encouraged us to examine in-depth AC effects in a model closer to the in vivo conditions. Results in rat brain slices showed that AC was effective in preventing the injury caused by H_2_O_2,_ although neuroprotection occurred at higher concentrations than in SH-SY5Y cells, as already reported [11,14,19]. AC also prevented ROS formation and lipid peroxidation. Catechins polyphenols are biological antioxidants with radical scavenging properties and among them, EGCG and ECG are the most potent, owing to their chemical structure characteristics such as the ortho-3′, 4′-dihydroxy- and the 4-keto, 3-hydroxyl or 4-keto and 5-hydroxyl-moieties [51]. AC mainly contains catechin and epicatechin, the less potent in terms of antioxidant activity, thus suggesting that neuroprotection probably depends also on other mechanisms, besides that of being a mere antioxidant.

ROS generation affects cellular antioxidant defences, induces oxidative damage to membrane lipids, cellular proteins and DNA, and it is crucial for the regulation of the main pathways of apoptosis mediated by mitochondria, death receptors and the endoplasmic reticulum [28]. The mitochondrial pathway of apoptosis is regulated by the Bcl-2 family of proteins, which consist of pro-apoptotic (Bax) and anti-apoptotic (Bcl-2) members [28]. Results showed that upon H_2_O_2_ challenge, there was increased formation of ROS and MDA formation, as well as an up-regulated Bax/Bcl-2 ratio and caspase-3-activation occurred in brain slices, suggesting the activation of the mitochondrial pathway. Interestingly, AC prevented the increase in ROS, MDA, Bax/Bcl-2 ratio, and pro-caspase-3-cleavage, indicating a protective role of the extract. This result agrees with reports describing the protective role of catechins and among these, EGCG is actually the most studied [39]. This polyphenol, in fact, can modulate multiple brain targets, such as many intracellular signalling pathways such as PKC, MAPK, and PI3K/Akt, survival and cell death genes (anti-apoptotic activity), can induce neurite growth, and stabilize the mitochondrial potential [51]. In particular, it has been reported that polyphenols of green tea protect PC-12 cells from H_2_O_2_-induced OS injury by enhancing cell survival and proliferation via the JNK signalling pathway [52]. Moreover, EGCG protects hippocampal neurons and neuroblastoma SH-SY5Y cells by suppressing ROS generation and by modulating the PI3K/Akt signalling cascade in order to decrease pro-apoptotic proteins such as Bax [48], in agreement with the present results.

To gain insights into these results, the principal signalling pathways altered by H_2_O_2_, such as p38, ERK1/2, and CaMKII, that also boost the activation of the first two, were analysed. Indeed, H_2_O_2_ is a well-established activator of the p38 MAPK signalling pathways [53], although the mechanism is not fully understood. Some evidence suggests that in mammalian cells the multifunctional antioxidant peroxiredoxins enzymes are required for H_2_O_2_-induced p38 activation [53]. However, at least in the present experimental conditions, p38 did not participate in the cascades activated by the H_2_O_2_ challenge. In addition, the latter is known to act also on Src-family protein tyrosine kinases, stimulating the activation of the MEK1/ERK1/2 signalling pathway [54], in agreement with the present findings. Activation of ERK1/2 was however only slightly reduced by AC, suggesting that its neuroprotective effects are not mediated by this signalling pathway, or that a longer period of incubation or a higher concentration of AC might be necessary to unveil its activity on ERK1/2.

CaMKII dysfunction and OS are both implicated in neurodegenerative diseases by causing dysregulation of calcium homeostasis and redox imbalance [55]. CaMKII is a ROS-sensitive signalling protein [56], as supported by the observation that mice in which CaMKII was made insensitive to ROS, were protected by diseases characterized by elevated OS [57]. The present results clearly showed that OS induced a significant rise in CaMKII activation. To explain the resulting apoptosis, we can hypothesize that increased phosphorylation of CaMKII might be responsible of endoplasmic reticulum (ER) stress, which depletes the calcium stores within the ER lumen, causing calcium increase in the cytoplasm [58], which in turn triggers apoptosis (Figure 9).

It is reasonable also to speculate that the increased cytoplasmatic calcium could be also taken up by mitochondria causing mitochondrial swelling and release of pro-apoptotic factors into the cytosol and apoptosis, as it is reported to occur under pathological conditions [28]. Moreover, SERCA2a protein is also sensitive to OS as free oxygen radicals directly inhibit ATPase activity by interfering with the ATP binding, thereby impairing the SR calcium pump rate [59,60]. This observation supports the present results, showing that the hydrogen peroxide challenge caused a reduction in SERCA2a expression. When rat brain slices were treated with AC the effects caused by OS on CaMKII and SERCA2a were totally or partially prevented, respectively. Taken together, these findings allow us to assume that the resulting neuroprotection might be a consequence of a direct effect on CaMKII activation. This hypothesis is supported by the observation that catechins directly interact with CaMKII, thus forming protein thiol adducts in vitro [61].

Considering the hypothesis of an altered gene expression as a consequence of the perturbation of the above-mentioned signalling pathways, a proteomic investigation on ACSF-, H_2_O_2_-, and H_2_O_2_+AC-treated rat brain slices were performed. The results suggested complex deregulation of the proteome profile brought about by OS-mediated injury, which however was partially prevented by AC treatment. Interestingly, the neuroprotective effects attributed to AC are consistent with its capability to rescue the abundance of fundamental proteins involved in synaptic vesicle formation and trafficking (i.e., Syngr3 and Syn1). Similarly, alterations in both proteins were observed in many neurological diseases [62,63]. Syngr3 is a transmembrane protein that localizes exclusively to synaptic vesicles that participates in the synaptic vesicles fusion process, although its role in such process is not yet fully elucidated [62,64]. On the other hand, Syn1 plays a role in the regulation of neuronal plasticity and synaptogenesis, including the regulation of synapse development, modulation of neurotransmitter release, and formation of nerve terminals [65,66]. The emerging picture is therefore consistent with the possibility that AC might represent a protective therapy targeting also synaptic dysfunctions, acting on signalling pathways that converge to the regulation of the synaptic vesicle release process, thus representing a safeguard against neurons excitability.

## 5. Conclusions

The research of effective treatments for ND represents a significant challenge. Natural compounds constitute an important source of neuroprotective agents by acting simultaneously on multiple targets. The understanding of their activity might drive the research toward the discovery of novel drugs for delaying the onset or the progression of ND [5,6]. The intake of phytochemicals on a regular basis might also boost the antioxidant system, thus increasing neuronal cell survival and improving physical and mental activity [38]. The present findings highlight interesting neuroprotective properties of AC, which prevented the formation of ROS and lipid peroxidation, as well as the changes in the expression of the main proteins involved in apoptosis and signalling pathways related to calcium homeostasis induced by OS. Additionally, AC prevented OS-induced changes in the proteome, including proteins belonging to the synaptic vesicle fusion apparatus, thus demonstrating to potentially impact also on synaptic dysfunctions, one of the major determinants of many ND.

## Figures and Tables

**Figure 1 antioxidants-11-00081-f001:**
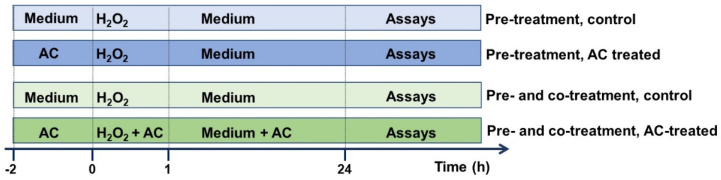
Neuroprotective effects of AC towards H_2_O_2_-induced injury in SH-SY5Y cells: experimental protocol used. To evaluate the potential of AC to prevent the injury, SH-SY5Y cells were treated with the extract (1–10 μg/mL) 2 h before (pre-treatment protocol) or 2 h before and during (pre- and co-treatment) cell injury (H_2_O_2_ 25 µM for 1 h, followed by 24 h with medium).

**Figure 2 antioxidants-11-00081-f002:**
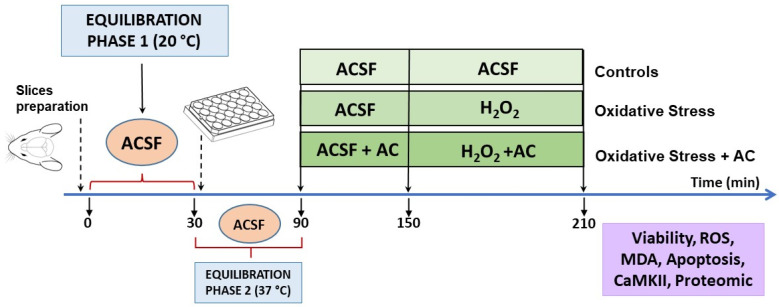
Neuroprotective effects of AC towards hydrogen peroxide-induced injury in rat brain cortical slices: scheme of the experimental protocol used.

**Figure 3 antioxidants-11-00081-f003:**
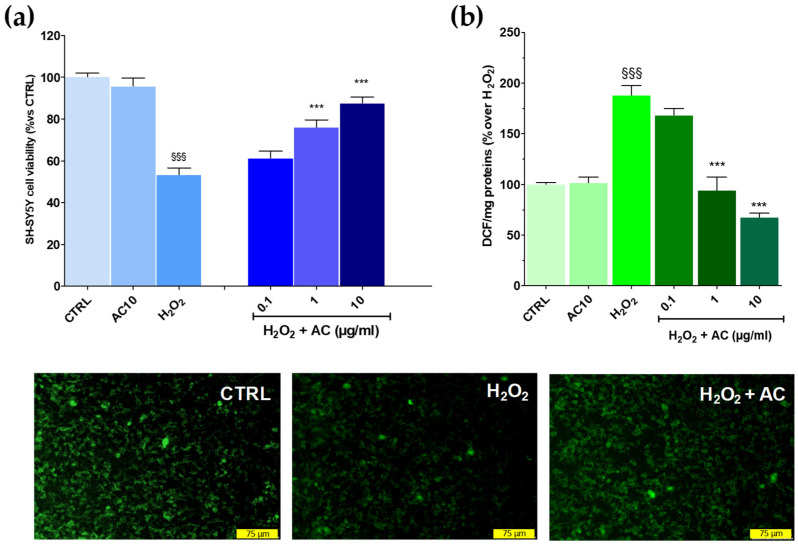
Effects of AC on oxidative stress (OS)-induced cytotoxicity, formation of ROS, and mitochondria membrane potential loss in SH-SY5Y cells. (**a**): Cell viability: cells were treated with AC (0.1–10 µg/mL) according to the pre- and post-treatment protocol described in Materials and Methods. OS was reproduced by using H_2_O_2_ (25 µM for 1 h + 24 h with medium). The effects of AC per se were tested at 10 µg/mL (AC10). (**b**): ROS formation as evaluated by the oxidation of DCF-DA to DCF on cells subjected to the AC pre- and post-treatment protocol and OS. Data are reported as means ± SEMs. §§§ *p* < 0.001, vs. untreated cells (CTRL); *** *p* < 0.001, vs. H_2_O_2_ (ANOVA and Bonferroni post-hoc test). (Lower panels): Representative fluorescence images of changes in the mitochondrial membrane potential, as assessed by Rhodamine 123 staining in cells subjected to the AC 10 µg/mL pre- and post-treatment protocol and OS. Each photograph was representative of at least three independent observations (scale bar 75 μm).

**Figure 4 antioxidants-11-00081-f004:**
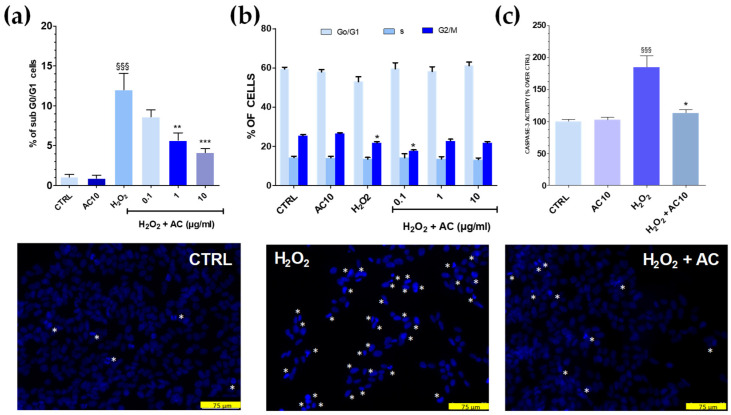
Effects of AC on OS-induced changes in SH-SY5Y cell cycle, DNA condensation, and caspase-3 activity. Cells were treated with AC (1–10 µg/mL) for 1 h before OS and for the following 24 h. OS was reproduced by using H_2_O_2_ (25 µM for 1 h + 24 h with medium). The effects of AC per se were tested at 10 µg/mL (AC10). (**a**): Cell cycle analysis: percent of cells in the sub G0/G1 (apoptotic) phase; (**b**) percent of cells in the G0/G1, S, or G2/M phase determined by flow cytometry after propidium iodide staining; (**c**) caspase-3 activity, assessed by using a caspase-3 specific substrate which releases the fluorescence probe AMC when activated. AC, in this case, was used at 10 µg/mL. Data are reported as means ± SEMs. §§§ *p* < 0.001, vs. untreated cells; * *p* < 0.05, ** *p* < 0.01, *** *p* < 0.001 vs. H_2_O_2_ (ANOVA and Bonferroni post-hoc test). (Lower panels): Apoptotic cells detected by DAPI staining. Asterisk indicates cells with fragmented nuclei and condensed DNA, considered apoptotic. Each photograph was representative of three independent observations (scale bar 75 μm).

**Figure 5 antioxidants-11-00081-f005:**
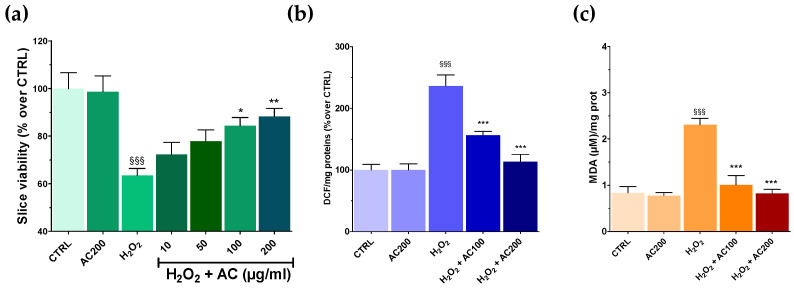
Effects of AC on OS-induced reduction in rat brain slices viability, formation of ROS, and malondialdehyde (MDA). Slices were incubated with artificial cerebrospinal fluid (ACSF) (controls) or ACSF + AC for 1 h. Afterward, AC was maintained and H_2_O_2_ (5 mM, 1 h) was added. The effects of AC per se were tested at 200 µg/mL (AC200). (**a**) Viability was assessed by MTT assay. (**b**) ROS was evaluated by the oxidation of DCF-DA to DCF. (**c**) Thiobarbituric acid reactive substances formed as a by-product of lipid peroxidation detected using thiobarbituric acid and reported as MDA formed. Data are shown as means ± SEMs. §§§ *p* < 0.001, vs. untreated slices (CTRL); * *p* < 0.05, ** *p* < 0.01, *** *p* < 0.001 vs. H_2_O_2_ (ANOVA followed by Bonferroni post hoc-test).

**Figure 6 antioxidants-11-00081-f006:**
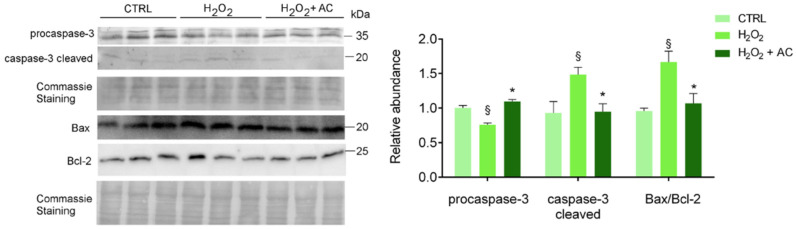
AC reverted the activation of H_2_O_2_-induced apoptotic proteins: WB analysis of rat brain slice lysates maintained in ACSF (CTRL), treated with H_2_O_2_, or with H_2_O_2_ in the presence of 200 µg/mL AC (H_2_O_2_ + AC). Antibodies to procaspase-3 (inactive isoform), caspase-3 cleaved (active isoform); the pro-apoptotic factor Bax and the anti-apoptotic factor Bcl-2 were used. The left panel depicts a representative WB along with the Coomassie blue staining of the membrane. Here and after, molecular mass standards (kDa) are reported on the right of each WB image. The right bar graph reports the densitometric analysis for the expression of proteins in the different samples (previously normalized to the optical density of the corresponding Coomassie-stained lane). Data are presented as mean ± SEMs. § *p* < 0.05 vs. CTRL, * *p* < 0.05 vs. H_2_O_2_ (ANOVA followed by Holm Sidak’s multiple comparison test).

**Figure 7 antioxidants-11-00081-f007:**
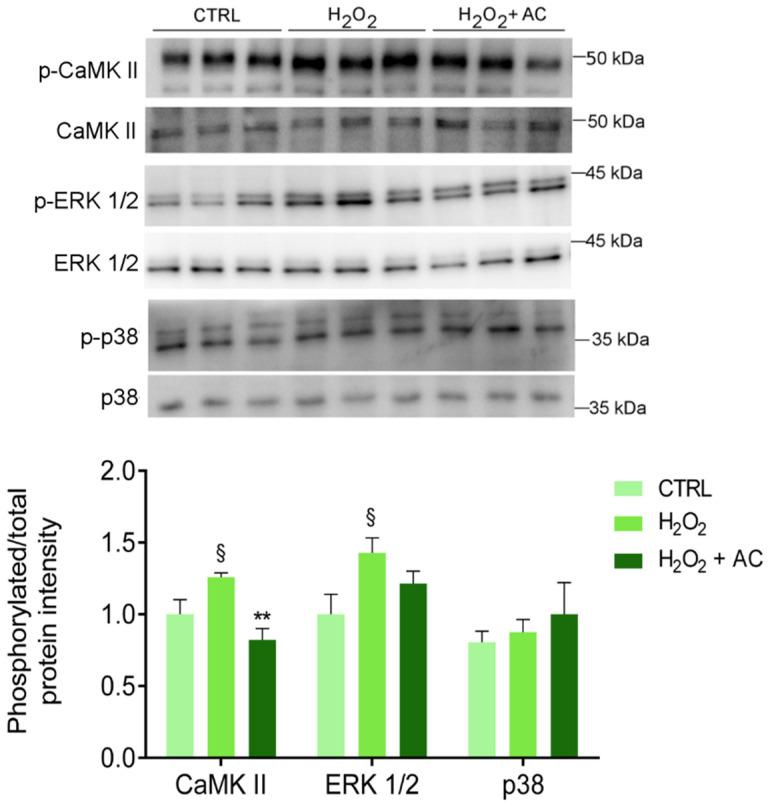
The H_2_O_2_-induced activation of CaMKII is rescued by AC (200 µg/mL) treatment. Protein lysates derived from rat brain slices (treated as described in Materials and Methods) were analysed by WB using antibodies against the phosphorylated isoforms of CaMKII, p38, and ERK 1/2 or their corresponding total protein. The upper panel shows a representative WB, while the bar diagram shows the ratio between p-CaMKII and CaMKII, p-p38, and p38, or p-ERK1/2 and ERK1/2 band intensity, respectively. § *p* < 0.05 vs. CTRL, ** *p* < 0.01 vs. H_2_O_2_ (ANOVA followed by Holm Sidak’s multiple comparison test).

**Figure 8 antioxidants-11-00081-f008:**
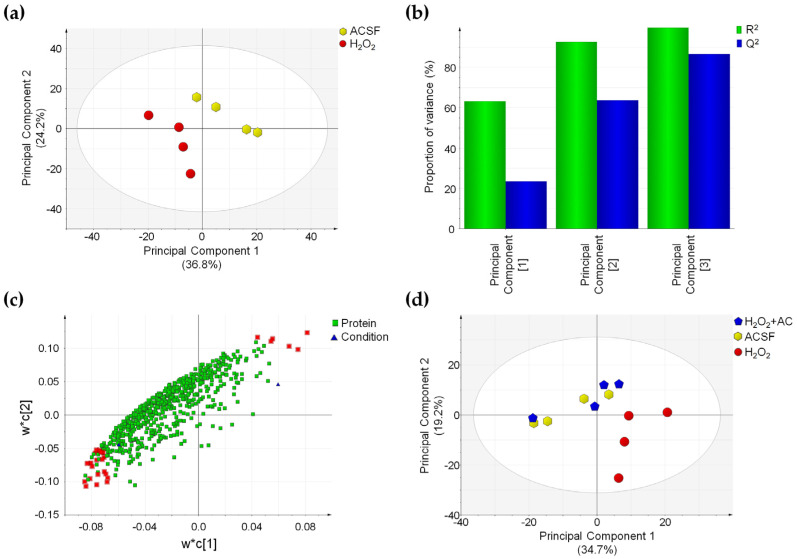
OS-induced proteome changes are reverted by AC (200 µg/mL) treatment. (**a**) PLS-DA score plot showing that ACSF- (controls, yellow hexagons) and H_2_O_2_-treated brain slices (red circles) are clustered in two distinct groups. (**b**) Summary of fit plot reporting the cumulative amount of explained data variance (green bars) and the cumulative predictive capability (blue bars) of the PLS-DA model when three principal components are used. (**c**) Loading scatter plot showing the 665 identified and quantified proteins reporting the 28 most important highlighted in red, which drive the separation between control and OS-injured rat brain slices. (**d**) PLS-DA score plot in which the three conditions are compared: a separation between H_2_O_2_-treated (red circles) and ACSF-treated (controls) brain slices (yellow hexagons) is appreciable, while those H_2_O_2_ + AC (blue pentagon) are clustered together with control brain slices. Label-free quantitative data were mean-centered and scaled to unit variance prior to analysis.

**Figure 9 antioxidants-11-00081-f009:**
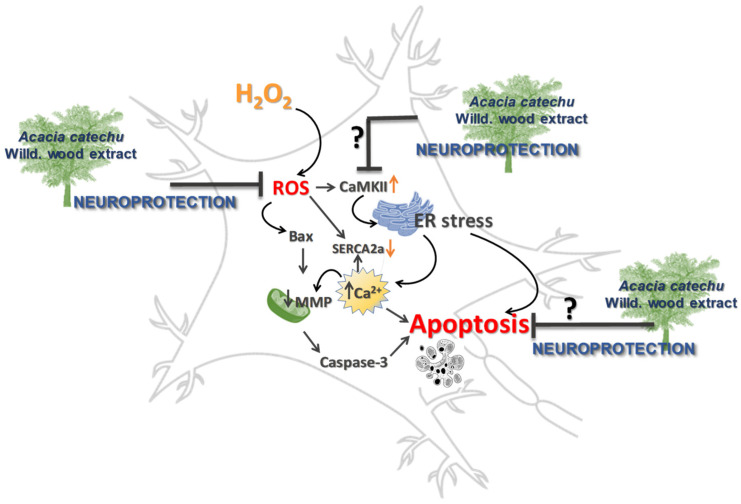
Neuroprotection afforded by AC, possible or hypothetical mechanisms.

## Data Availability

MS proteomics data have been deposited via the PRIDE partner repository with the dataset identifier PXD029239.

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
