# Peer review of "Acacia catechu Willd. Extract Protects Neuronal Cells from Oxidative Stress-Induced Damage"

_antioxidants, 2021, doi:10.3390/antiox11010081_

Round 1
Reviewer 1 Report
The authors present a potentially interesting article on the anti-oxidative properties of an extract from Acacia Catechu Willd. in immortalized neuronal cells and rat brain. The extract is proposed as a nutraceutical potentially protecting against neurodegenerative diseases. The results presented in the article support this hypothesis, however some major limitation suggest a more cautious interpretation:
1. In general anti-oxidants are known to turn into oxidants depending on their enviroment, i.e. the actual conditions of the cells. Therefore, antioxidants may rather damage instead of protecting cells, once inflammation is already ongoing. Of course H2O2 exposure may mimick the oxidative stress as expected during inflammation, even though in an inflammatory state a more chronic exposure may be expected. Another important aspect of inflammation may be the increased acidity of the cells. I would like to ask the authors if they noted changes in the pH of the medium. Which effects may a change in pH have of the anti-oxidative properties of the extract? Could some cell-culture experiments be performed with slightly different pH-values of the medium (e.g. pH 7,0, 7,4, 7,8)?
2. In my opinion the post-treatment experiments are not really post-treatments, since the extract was present already before H2O2. This is similar to my comment above, since a pretretment with H2O2 alone followed by AC better reflects the situation where a therapy is started once a damage is already present. Why was this strategy not tested?
Author Response
Ref 1
The authors present a potentially interesting article on the anti-oxidative properties of an extract from Acacia Catechu Willd. in immortalized neuronal cells and rat brain. The extract is proposed as a nutraceutical potentially protecting against neurodegenerative diseases. The results presented in the article support this hypothesis, however some major limitation suggest a more cautious interpretation:
- In general anti-oxidants are known to turn into oxidants depending on their enviroment, i.e. the actual conditions of the cells. Therefore, antioxidants may rather damage instead of protecting cells, once inflammation is already ongoing. Of course H2O2 exposure may mimick the oxidative stress as expected during inflammation, even though in an inflammatory state a more chronic exposure may be expected. Another important aspect of inflammation may be the increased acidity of the cells. I would like to ask the authors if they noted changes in the pH of the medium. Which effects may a change in pH have of the anti-oxidative properties of the extract? Could some cell-culture experiments be performed with slightly different pH-values of the medium (e.g. pH 7,0, 7,4, 7,8)?
We agree with the Referee that Inflammation and oxidative stress are closely related and tightly linked pathophysiological processes, although the understanding of which one trigger the other is still debated (see for ex. PMID 26881031). Hydrogen peroxide challenge is a widely used “acute” experimental model, which may not be relevant to the protracted degeneration observed in patients characterized by a low-grade chronic inflammation and oxidative stress. Nevertheless, it has greatly increased our knowledge of the biochemical mechanisms of neuronal cell death in neurodegenerative diseases and is still very helpful to make a preliminary screening of potentially neuroprotective drugs as well as to characterize their mechanism of action. In the present study, AC was proven to be neuroprotective and if this activity is still retained also in in vitro “chronic” oxidative stress conditions or in vivo models is a matter of future studies. Regarding the changes in pH of the medium, this was not observed in SH-SY5Y experiments. Cell culture medium contains phenol red that has a transition point of 7.5 (acid side to yellow / alkaline side to red). Hydrogen peroxide was added to cells for 1h, and then it was removed and replaced with fresh medium for 24h. After this time, we did not noticed changes in medium colour, but we cannot however exclude that modest changes in pH occurred. In brain slices, the ACSF used is a buffered solution. We have however checked pH at the end of the experiments, and this was unchanged in both controls- and H2O2-treated samples. Finally, data from the literature reports that antioxidant activities of catechins are high and constant within the pH range of 6-12 and these are affected only in strong acid/basic conditions (PMID 11272815). Taken all together, we can reasonably hypothesize that antioxidant capability of AC is not affected in our experimental conditions.
- In my opinion the post-treatment experiments are not really post-treatments, since the extract was present already before H2O2. This is similar to my comment above, since a pretretment with H2O2 alone followed by AC better reflects the situation where a therapy is started once a damage is already present. Why was this strategy not tested?
We fully agree with the Referee that the used term “post-treatment” is misleading. This has been changed into “co-treatment”, which better resemble the model used. We however partially agree whit Referee that the use of AC after the oxidative stress would better resemble the in vivo condition and thus its use as a “curative” drug. In this regard, we should consider that this approach is more useful in “acute” ND such as during/after ischemic stroke. In this case is mandatory to rescue damaged tissue as much as possible and natural compounds are actually effective when given after the injury (see for ex. PMID 21127495). On the other hand, chronic ND are characterized by a loss of the neurons occurring over a long period. The fact that pathological changes at molecular and cellular levels precede the clinical onset by several years, underscores a pressing need for the initiation of interventions before the emergence of neurological symptoms. Hence, the prevention of such preclinical progression of neurodegeneration plays a vital role in the successful translation of data from studies using animal and cellular models. This is why we should be able to develop therapies that prevent/slow or halt the neurodegeneration and for this we decided to use a pre- co-treatment protocols. We realize that this was probably not sufficiently clear, and thus we have better elucidated it in the Introduction (see lines 59-63 of revised version) and in point 2.2.3 of Mat&Met (lines 132-137 of revised version).
Reviewer 2 Report
The study titled "Acacia Catechin Willd. extracts protects neuronal cells from oxidative in-induced damage" by Chiaino et al. describes experiments that test whether plant extract from Acacia (AC)reduce the effects of oxidative stress in cell lines (SH-SY5Y) and rat brain slices. The authors employ multiple complmentary assays to assess cellular damage. They also perfrom mass spec analyses to assess, which proteins are protected by AC treatment.
The manuscript is overall clear and easy to follow. Some English editing would improve the manuscript.
That data is overall convincing and support the conclusions.
The Discussion seems in part quite speculative and could be shortened to focus on those aspects directly addressed in this study.
The methods are described effectively and in sufficent detail. Figures 1 and 3 nicely illustrate the experimental approach.
Some experiments (e.g. Figure 3) do no include the AC only treatments, which should be added for each experiment.
The caspase-3 Western blot in Figure 6 is of porr quality and needs to me replaced a a higher quality blot.
AS a limitation, only hydrogenperoxide but no other other oxidative stress treatment is used.
A conceptual limitation is the lack of noelty since AC contains many previously described anti-oxidants and the results presented here are thus not overly surprising.
Author Response
Ref 2
The study titled "Acacia Catechin Willd. extracts protects neuronal cells from oxidative in-induced damage" by Chiaino et al. describes experiments that test whether plant extract from Acacia (AC)reduce the effects of oxidative stress in cell lines (SH-SY5Y) and rat brain slices. The authors employ multiple complmentary assays to assess cellular damage. They also perfrom mass spec analyses to assess, which proteins are protected by AC treatment.
The manuscript is overall clear and easy to follow. Some English editing would improve the manuscript; That data is overall convincing and support the conclusions; The Discussion seems in part quite speculative and could be shortened to focus on those aspects directly addressed in this study; The methods are described effectively and in sufficent detail. Figures 1 and 3 nicely illustrate the experimental approach; Some experiments (e.g. Figure 3) do no include the AC only treatments, which should be added for each experiment; The caspase-3 Western blot in Figure 6 is of porr quality and needs to me replaced a a higher quality blot.; AS a limitation, only hydrogenperoxide but no other other oxidative stress treatment is used.; A conceptual limitation is the lack of noelty since AC contains many previously described anti-oxidants and the results presented here are thus not overly surprising.
We thank the Referee for its suggestion. The following changes have been performed:
-Speculative parts in discussion have been deleted;
-AC per se effects have been added;
-Caspase 3 WB has been replaced. In this regard, however, several papers report that in contrast to the proenzyme, once activated caspase-3 is difficult to detect due to its rapid degradation (e.g., https://doi.org/10.1038/sj.cdd.4401360). For this reason, due to the low intensity of the bands relative to the active cleaved caspase 3 and to support the data that AC reduced the activation of caspase-3, we decided to add the WB relative to the procaspase-3 isoform not cleaved, showing also in the graph the relative amount of procaspase isoforms in the different conditions.
-The reason of using hydrogen peroxide challenge to resemble OS is that this approach is feasible in both cells and brain slices and, being widely used, allows us also a better comparison of the results with data from the literature.
Regarding the lack of novelty, in our opinion the interesting aspect of the present study is that AC bark used contains high amount of of catechins (10-20 times higher that green tea). The results highlighted a very high profile of activity of the extract, occurring at concentrations of at least one or two orders of magnitude higher than other well-known catechin such as EGCG. The activity at low concentration is interesting also in consideration of the hormetic, bell shaped behaviour of catechins, characterized by an “efficacy window” of neuroprotective activity in the low micromolar range, whereas they become pro-oxidant and/or pro-apoptotic for higher (>10-50 µM) concentrations. This is also the case of AC, which exert pro-oxidant activity at a concentration 25 times higher than that demonstrated to protect neurones. The wider limit between protectant- and damaging-concentrations constitutes an added value of AC. Moreover, we have also performed for the first time a proteomic analysis, which highlighted that AC prevented the hydrogen peroxide-induced changes in the proteome, including proteins belonging to the synaptic vesicle fusion apparatus, further supporting the possibility of AC as a nutraceutical useful in preventing neurodegenerative diseases.
Reviewer 3 Report
The main concern is that the experimental design lacks a real neuroprotective and well-proven drug to compare the denominated AC that seems to be catechins. On line 120, Figure 1. The pre- and post-treatment with AC, in fact, is a “continuous treatment” because AC was never removed, if this was the case, as stated on lines 362-363 “Therefore, the pre- and post-treatment protocol was then selected to further assess neuroprotective effects of AC”, as well as in discussion (line 587) then all the conclusions along the paper are misinterpreted because the observed effects would be in fact an extract that scavenges H2O2, such as many plant extract does (Fernando & Soysa, 2015). Furthermore, all along through the text, the word “reverted” should be substituted for “preventing” since the AC as an H2O2 scavenger never let the H2O2, in its full concentration to act, if this is the case, figure 9 should be edited to show the blocking effect of H2O2 by AC in only single neuroprotection point that is the scavenging of H2O2 .
Despite the above comments, the manuscript is written and suitable to be published.
Please check
Line 359: supllementary
Line 364 you mean innocuous instead or ineffective?
Please define the acronym OS (oxidative stress) only once, the first time, thereafter, use OS.
In microscopy images the scale bars are barely visible, please bold them
Line 396: cells
Line 426: neurorptective
Line 428: Byin
References
Fernando, C. D., & Soysa, P. (2015). Optimized enzymatic colorimetric assay for determination of hydrogen peroxide (H2O2) scavenging activity of plant extracts. MethodsX, 2, 283–291. https://doi.org/10.1016/j.mex.2015.05.001
Author Response
Ref 3
The main concern is that the experimental design lacks a real neuroprotective and well-proven drug to compare the denominated AC that seems to be catechins. On line 120, Figure 1. The pre- and post-treatment with AC, in fact, is a “continuous treatment” because AC was never removed, if this was the case, as stated on lines 362-363 “Therefore, the pre- and post-treatment protocol was then selected to further assess neuroprotective effects of AC”, as well as in discussion (line 587) then all the conclusions along the paper are misinterpreted because the observed effects would be in fact an extract that scavenges H2O2, such as many plant extract does (Fernando & Soysa, 2015). Furthermore, all along through the text, the word “reverted” should be substituted for “preventing” since the AC as an H2O2 scavenger never let the H2O2, in its full concentration to act, if this is the case, figure 9 should be edited to show the blocking effect of H2O2 by AC in only single neuroprotection point that is the scavenging of H2O2 . Despite the above comments, the manuscript is written and suitable to be published.
We agree with the Referee that a positive control would be useful. AC is however a “complex” mixture of bioactive compounds and in our opinion the comparison of its effects with those elicited by a “single” compound would not have been appropriate.
As far as the second point raised, we fully agree that the term “post-treatment” is misleading and it has been changed into “co-treatment”. The use of this model is based on the idea that as in chronic ND pathological changes at molecular and cellular levels precede the clinical onset by several years, the prevention of such preclinical progression of neurodegeneration plays a key role. Thanks to the point raised by the Referee 3 and 1, we realize that this was probably not sufficiently clear, and thus we have better elucidated it in the Introduction (see lines 59-63 of revised version) and in point 2.2.3 of Mat&Met (lines 132-137 of revised version). Moreover, all the misinterpretations about “prevention of the effects” have been properly amended and in Figure 9, the possible direct effects of AC of CaMKII and apoptosis has been now reported as only hypothethical.
Minor points:
Line 364 you mean innocuous instead or ineffective? We meant that it was inactive, the misleading word has been replaced.
Please define the acronym OS (oxidative stress) only once, the first time, thereafter, use OS. Appropriate changes have been performed.
In microscopy images the scale bars are barely visible, please bold them. Scale bar have been bolded in both figures.
Line 396: cells. Amended
Line 426: neurorptective. Amended
Line 428: Byin. Amended.
Round 2
Reviewer 2 Report
The revisions greatly strengthened the manuscript, which seems ready for publication now.